Reproducibility of SNV-calling in multiple sequencing runs from single tumors

Derryberry Dakota Z. 1 dakotaz@utexas.edu
Cowperthwaite Matthew C. 2 3
Wilke Claus O. 4 5
1 Cell and Molecular Biology, The University of Texas at Austin , Austin, TX , United States
2 NeuroTexas Institute Research Foundation , Austin, TX , United States
3 Center for Systems and Synthetic Biology, The University of Texas at Austin , Austin, TX , United States
4 Integrative Biology, The University of Texas at Austin , Austin, TX , United States
5 Center for Computational Biology and Bioinformatics, The University of Texas at Austin , Austin, TX , United States
Zhao Min
Electronic publication date: 2016 Jan 4
Publication date: 2016
Volume: 4
Electronic Location ID: e1508
Received 2015 Jun 5; Accepted 2015 Nov 25
Copyright: ©2016 Derryberry et al.
Copyright year: 2016
Copyright holder: Derryberry et al.
License: This is an open access article distributed under the terms of the Creative Commons Attribution License, which permits unrestricted use, distribution, reproduction and adaptation in any medium and for any purpose provided that it is properly attributed. For attribution, the original author(s), title, publication source (PeerJ) and either DOI or URL of the article must be cited.
License URL: https://creativecommons.org/licenses/by/4.0/

Keywords: Benchmarking, Cancer, Exome sequencing, SNV-calling, TCGA, Glioblastoma

Funding: NSF Cooperative Agreement DBI-0939454 St. David’s Hospital’s NeuroTexas Institute Research Foundation The Texas Advanced Computing Center This work was supported in part by NSF Cooperative Agreement No. DBI-0939454 (BEACON Center), and by St. David’s Hospital’s NeuroTexas Institute Research Foundation. The Texas Advanced Computing Center provided high performance computing resources. The funders had no role in study design, data collection and analysis, decision to publish, or preparation of the manuscript.

==============================
We examined 55 technical sequencing replicates of Glioblastoma multiforme (GBM) tumors from The Cancer Genome Atlas (TCGA) to ascertain the degree of repeatability in calling single-nucleotide variants (SNVs). We used the same mutation-calling pipeline on all pairs of samples, and we measured the extent of the overlap between two replicates; that is, how many specific point mutations were found in both replicates. We further tested whether additional filtering increased or decreased the size of the overlap. We found that about half of the putative mutations identified in one sequencing run of a given sample were also identified in the second, and that this percentage remained steady throughout orders of magnitude of variation in the total number of mutations identified (from 23 to 10,966). We further found that using filtering after SNV-calling removed the overlap completely. We concluded that there is variation in the frequency of mutations in GBMs, and that while some filtering approaches preferentially removed putative mutations found in only one replicate, others removed a large fraction of putative mutations found in both.

Introduction

Glioblastoma multiforme (GBM) is the most common and deadly primary brain tumor, with a median survival time of 14 months and a 5-year survival rate of 5%. Prognosis for patients with this disease remains poor despite significant research investment, due to the difficulty of surgical resection and the limited number of effective chemotherapeutics (Wilson et al., 2014). Recently, research to improve treatments for patients with this devastating disease has focused on the idea of precision medicine, that is using large-scale genomics data to discover how the disease arises and progresses, and how to stop it. The past six years have seen an explosion of data in cancer genomics, an effort led by TCGA, an archive of publicly-available data that includes sequencing of paired tumor-normal samples from individual patients for thousands of tumors (The Cancer Genome Atlas Research Network, 2008; Brennan et al., 2013), including over 500 GBMs. With these and other similar data (Parsons et al., 2008), researchers have discovered genes and pathways mutated in GBM (Cerami et al., 2010), different GBM subtypes (Verhaak et al., 2010), and a variety of computational models to find GBM driver mutations (Gevaert & Plevritis, 2013). Despite widespread use of the data and adoption of the methods, however, efforts to benchmark the data—to assess the quality and repeatability of these and subsequent analyses—have lagged or been non-existent. Here, our goal is to begin to develop robust methods for benchmarking next-generation cancer sequencing data and analysis.

It is well known that sequencing and variant-calling pipelines are not error-free. For example, different pipelines for calling single-nucleotide variants (SNVs) can return different results on the same data (Yu & Sun, 2013). Given the heterogeneity in cancer genomes (Kumar et al., 2014; Friedmann-Morvinski, 2014), and the presence of functional low-frequency variants in GBM (Nishikawa et al., 2004), the signal-to-noise ratio in the TCGA dataset may be particularly low. Yet despite widespread use of this dataset, and significant monetary investment in collecting and analyzing the data, we know little about how to maximize the quality of the sequence data and SNV-calls. One way to address this question would be to analyze technical replicates of sequencing data (Robasky et al., 2014). Here, we addressed one aspect of this question, by asking whether the same SNV-calling pipeline will return comparable results on two sequencing runs from the same tissue. And further, we asked whether added filtering after SNV-calling increases or decreases the degree of similarity among replicates.

We answered these questions using TCGA data from 55 GBM tumors that were sequenced twice, once each with (i) the standard whole-genome sequencing (WGS) protocol, and (ii) an additional amplification step before library prep, which we refer to as the whole-genome amplification (WGA) protocol. For each of these 55 technical replicate pairs, we compared the somatic variants found in the WGS and WGA replicates, before and after analyzing the variants using various SNV-filtering approaches. We found significant overlap (around 50%) between technical replicates, but also significant differences. As expected, the additional amplification step in the WGA protocol versus the WGS protocol added some putative SNVs to the sample, so that on average these replicates had (i) more putative SNVs, and (ii) a smaller percentage overlap between replicates. We found that the number of SNVs in the WGS replicates varied by orders of magnitude, from 110 to 8,192. Contrary to expectations, the percentage overlap between technical replicates did not decrease with increasing numbers of putative SNVs, suggesting real variation in mutation frequency (the absolute number of SNVs found in a tumor) between tumors. This result may be in part due to known mutational hotspots in some tumors (Wang et al., 2008). Our attempt to use SNV filtering, either via SomaticSniper (Larson et al., 2012) or via custom-built filters, to increase the similarity between technical replicates, was unsuccessful: filtering removed almost the entirety of the overlap between the replicates.

Methods

Data and back-end processing

All sequence data came from The Cancer Genome Atlas (TCGA) Research Network’s Glioblastoma multiforme (GBM) data set (The Cancer Genome Atlas Research Network, 2008). We downloaded three BAM files for each of 55 patients using CGHub (University of California Santa Cruz, 2014). For each patient, data consisted of one BAM file taken from blood DNA, and two BAM files from tumor DNA, one for each technical replicate. In each case, the only difference in data collection for the two sets of tumor DNA was whether or not an amplification step was performed prior to building a library (The Cancer Genome Atlas Research Network, 2008).

We created a pipeline for backend processing of all TCGA BAM files, summarized in Fig. 1, with commands given in Table S1. The custom python code to connect the pipeline is available in a public git repository (https://github.com/clauswilke/GBM˙genomics). Our pipeline first regenerated fastq files (original reads) from the TCGA BAM files, which were aligned to hg18, using picard (Broad Institute, 2014). Next, we used BWA (Li & Durbin, 2009) to align the fastq files to hg19, and samtools (Li et al., 2009) to sort, index, and de-duplicate the new BAM file. We used GATK (McKenna et al., 2010) to do indel realignment and base recalibration, according to the standard best practices for genomics data (GATK Development Team, 2015). Finally, we predicted somatic variants with SomaticSniper (Larson et al., 2012), with the output given in VCF format.

Figure 1 Data processing pipeline.

For each of 55 patients, we began with a C484 tumor BAM file (WGS), a C282 tumor BAM file (WGA), and a normal BAM file, all aligned to hg18. For each BAM file, we used picard to regenerate fastq files, bwa to realign the fastq files to hg19, and GATK to recalibrate bases and indels. We used SomaticSniper to call somatic mutations (differences between the tumor and normal sequences) for each replicate. When we had a VCF for each replicate, we calculated the overlap between the two lists as the number of individual mutations which appeared in both replicates.

For all alignments, average read coverage was 30X, with a low of 5X and a high of 60X (stdev = 15.95898). The percentage of mapped reads in our alignments was universally high (mean = 98.03545, stdev = 3.257872), with only two samples below 90% (68% and 74%), and only 15 samples below 98%. We found no correlation between the read coverage or the percentage of mapped reads and the total number of SNVs called (Figs. S2 and S3).

Filtering and data analysis

After generating a VCF file with all of the putative somatic variants for each replicate of each sample, we used custom python code (available in public git repository) to list the putative SNVs in each VCF, and to calculate the overlap between technical replicates. We then filtered the lists of putative SNVs according to the eight filters described in Table 2, using a combination of command line options and custom python code (available in a public git repository). Each of the eight filters was run independently on the whole data set. The eight filters were enacted as follows:

GATK: The GATK quality score is automatically generated by GATK; GATK recommends discarding all putative mutations with a quality score below 40, which we did using the command line option -q 40 when we called SomaticSniper (for the exact command, see Table 2). We did not at any point consider putative mutations removed by this filter, and did not consider its individual action.

SS: The SomaticScore is a similar metric calculated by SomaticSniper. As recommended, we removed from consideration all putative mutations with a SomaticScore below 40 by using the command line option -Q 40 when we called SomaticSniper (for the exact command, see Table 2). We did not at any point consider putative mutations removed by this filter, and did not consider its individual action.

VAQ: The Variant Allele Quality (VAQ) score calculated by SomaticSniper is a third measure of this type. SomaticSniper recommends discarding putative mutations with a VAQ below 40, which we accomplished using a custom python script available in the public git repository. This recommendation is discussed in Results and Discussion.

LOH: It is general (but not universal) practice to disregard Loss of Heterozygosity (LOH) in large scale genomics, because LOH too easily results from sequencing errors. We excluded LOH variants using a custom python script available in the public git repository. This recommendation is discussed in the Results and Discussion.

10bp-SNV: It is universal or near universal practice to exclude variants within 10 bp of another putative somatic variant, because clusters of putative mutations often indicate an error in reads or sequence alignment. We excluded 10bp-SNV variants using a custom python script available in the public git repository. This recommendation is discussed in the Results and Discussion.

10bp-INDEL: It is universal or near universal practice to exclude variants within 10 bp of a putative indel, because clusters of putative mutations often indicate an error in reads or sequence alignment. We excluded 10bp-INDEL variants using a custom python script available in the public git repository. This recommendation is discussed in the Results and Discussion.

dbSNP: It is universal or near universal practice to exclude variants that overlap with dbSNP, because the overlap often indicates an amplification error and not a true somatic variant. We excluded dbSNP variants using a custom python script available in the public git repository. This recommendation is discussed in the Results and Discussion.

<10%: It is universal or near universal practice to exclude variants when the alternate allele coverage is less than 10%, because the low coverage of the alternate allele often indicates sequencing error. We excluded <10% variants using a custom python script available in the public git repository. This recommendation is discussed in the Results and Discussion.

We used the same or substantially similar python scripts to calculate and compare the overlap between technical replicates before and after filtering. These scripts are also available in the public git repository. We plotted all data and did all statistics with standard plotting and statistics functions in R (R Core Team, 2014). This code is also available in the public git repository.

Results

How similar are two replicate sequencing runs of the same tumor?

DNA sequencing is not error-free. Error is introduced by mis-called bases in sequencing runs and by mis-aligned bases during sequence analysis (Wall et al., 2014). Loss of heterozygosity may be a real feature of the data, or an amplification artifact that occurs when only one allele of a polymorphic site is amplified. Cancer DNA is highly heterogenous, which makes for an additional source of error: a polymorphic mutation is defined as a mutation that is not fixed and therefore is not present in all tumor cells. Such mutations may or may not be represented at high enough frequency in a particular tumor specimen to be identified with next-gen sequencing. Alternatively, a polymorphic mutation may appear to be fixed when present at very high frequency in a tumor specimen. In general, we would like to know how often these sorts of errors occur. One way of investigating this question would be to use an orthogonal technique such as PCR to verify each individual mutation. However, this method is expensive and time consuming. A cheaper alternative would be to sequence the tumor multiple times and to look at the similarity between replicates. Theoretically, any fixed mutation will appear in all replicates, while errors due to (i) sequencing errors, (ii) amplification errors, or (iii) alignment errors will not (Polymorphic mutations would be present in some but not all samples, so this method does not address the difficulty of calling low-frequency variants in tumors.). The multiple-sequencing approach is used in most biological sequencing experiments, but not generally in cancer genomics, presumably due to unavailability of additional pathology specimens and the expense of sequencing multiple replicates for each of the hundreds of samples necessary for cancer genomics research. Nevertheless, even if researchers cannot verify every mutation or sequence multiple replicates for each tumor, it would be useful to know what percentage of called mutations would be likely to appear in additional sequencing replicates.

TCGA’s GBM data set includes 2 technical replicates for each of 55 tumors. In this case, the technical replicates are not identical. One protocol included an additional amplification step (The Cancer Genome Atlas Research Network, 2008), and we refer to this replicate as the WGA (whole-genome amplification) replicate, and the other as the WGS (whole-genome sequencing) replicate. Despite the difference in sequencing protocol, we would theoretically expect any fixed mutation to appear in both replicates, while polymorphic mutations and sequencing error might appear in only one. Consequently, those putative mutations appearing in both replicates are more likely to be real somatic variants than those found in only one replicate. We further hypothesized that the WGA samples would have a greater number of amplification errors, and thus more putative SNVs per sample, than the corresponding WGS samples.

For each patient (n = 55), we called mutations in both technical replicates and in the patient’s blood sample, using the same computational pipeline (see Fig. 1 and ‘Methods’): We downloaded TCGA BAM files with CGHub (University of California Santa Cruz, 2014), re-generated fastq files with picard (Broad Institute, 2014), re-aligned the fastq files to hg19 with bwa (Li & Durbin, 2009), performed indel re-alignment and base recalibration with GATK (McKenna et al., 2010), and finally called somatic mutations with SomaticSniper (Larson et al., 2012). We then compared the VCFs produced by SomaticSniper for each of the two technical replicates, and calculated the number of somatic mutations called in each replicate and the number of individual somatic mutations called in both replicates (hereafter, the overlap, see Fig. 1). We further calculated the percentage of mutations in each replicate that occurred in the overlap between the two.

There were an average of 844 putative mutations in the WGS replicates and 1,531 putative mutations in the WGA replicates (Table 1). Across all samples, the number of mutations in a given WGS replicate was correlated with the number of mutations in its corresponding WGA replicate (Spearman ρ = 0.42, S = 16142, P = 0.002, Fig. 2). As expected, for each sample the WGA replicate (with the additional amplification step) had slightly more mutations overall, with a slightly smaller percentage appearing in the overlap (Figs. 2 and 3). We found that the percent overlap between the two samples, calculated as WGA∩WGS∕WGS for WGS replicates and WGA∩WGS∕WGA for WGA replicates, varied widely, from 1%–74%, but was fairly evenly distributed around the average of 31% in WGA replicates and 44% in WGS replicates (Table 1). As expected, the distribution of the percentage overlap was narrower and taller in the WGS replicates, because on the whole the WGA samples had more amplification errors than the WGS samples (Fig. S1).

Table 1 Summary Statistics.

This table describes the results of our data processing pipeline across samples and pairs of replicates, in terms of the number of putative mutations found, length of the overlap between replicates, number of mutations removed (per sample) by each filter, and percent of the overlap removed (per pair of replicates) by each filter.

Quantity	Average	Median	Min	Max	Stdev	
No. mutations, WGS	844	328	110	8,192	1,306	
No. mutations, WGA	1,531	694	23	10,966	2,230	
% mutations in overlap, WGS	31%	31%	1%	74%	20%	
% mutations in overlap, WGA	44%	45%	3%	71%	13%	
No. putative mutations removed, VAQ	309	164	17	11,439	1,372	
No. putative mutations removed, LOH	539	169	3	8,538	1,051	
No. putative mutations removed, 10bp-SNV	46	19	0	1,515	115	
No. putative mutations removed, 10bp-INDEL	28	16	0	535	45	
No. putative mutations removed, dbSNP	16	8	0	348	33	
No. putative mutations removed, <10%	1	1	1	36	3	
% overlap removed, VAQ	53%	57%	0%	100%	28%	
% overlap removed, LOH	51%	52%	0%	99%	29%	
% overlap removed, 10bp-SNV	3%	2%	0%	20%	3%	
% overlap removed, 10bp-INDEL	4%	4%	0%	22%	4%	
% overlap removed, dbSNP	3%	2%	0%	47%	7%	

Table 2 SNV prediction filters.

This table shows the various methods (filters) used to predict which differences found in tumor alignments relative to blood alignments are real somatic variants, as opposed to sequencing errors or other variants.

Filter	Software	Purpose	
GATK	GATK	Removes putative SNVs with GATK quality scores less than 40 (as part of the GATK processing, with indel realignment and base recalibration)	
SS	SomaticSniper	Removes putative SNVs with a SomaticScore less than 40	
VAQ	SomaticSniper	Removes putative SNVs with SomaticSniper Varaint Allele Quality scores less than 20	
LOH	SomaticSniper, python	Removes putative SNVs that are identified as loss of heterozygosity	
10bp-SNV	python	Removes putative SNVs located within a 10 bp window of any other putative SNV	
10bp-INDEL	python	Removes putative SNVs located within a 10 bp window of indels	
dbSNP	python	Removes putative SNVs that overlap with dbSNP coverage	
<10%	python	Removes putative SNVs if, in the tumor data, the percentage of reads covering the site with the alternate allele is less than 10%	

Figure 2 Number of putative SNVs in WGS versus WGA, as called by SomaticSniper before filtering.

Each point represents data for a single patient. The line is y = x, so points falling below the line agree with the hypothesis that an additional amplification step produces more sequencing errors in a sample. The number of mutations found in one replicate correlates with the number of mutations found in the other replicate (Spearman ρ = 0.42, S = 16142, P = 0.002).

Figure 3 Number of putative SNVs per sample does not correlate with the number of putative SNVs recoverable in both replicates.

The percentage of putative SNVs in a given sample that are in the overlap between replicates is not correlated to the number of mutations in that sample (Spearman ρ = 0.05, S = 29268, P = 0.68). We calculated the percent overlap in two ways: with reference to the total number of putative SNVs in the WGS sample (green) and with reference to the total number of putative SNVs in the WGA sample (orange). The correlation was calculated with respect to WGA.

Although the numbers of putative mutations in WGS and WGA samples were correlated, the exact number of putative mutations in each sample varied by orders of magnitude (Table 1). It is known that different cancers mutate at different rates: some pediatric cancers have very few mutations (Knudson, 1971; Chen et al., 2015), while some adult tumors show a mutator phenotype leading to vastly increased numbers of mutations, usually resulting from errors in DNA repair pathways (Loeb, 2011). GBM specifically is thought to have a relatively low mutation rate (Parsons et al., 2008; Brennan et al., 2013), and while some of our samples had low mutation frequencies in line with this theory (29 out of 110 samples had a mutation frequency within a factor of 2 of the reported 3 mutations per Mbp genome), several samples also had mutation frequencies an order of magnitude greater (24 out of 110 samples had a mutation frequency greater than 30 mutations per Mbp genome). Possible explanations are a degraded DNA sample, significant alignment error, or otherwise bad data. If one of these were the case, we would expect the percentage of the overlap between replicates (a measure of data quality) to decrease with the overall number of putative mutations. However, we found no significant correlation (Spearman ρ = 0.05, S = 29268, P = 0.68) between the number of putative mutations in the WGS replicate and the percentage of those mutations that were in the overlap between replicates (Fig. 3). Thus, our data suggest that some samples may simply have a higher mutation frequency than others, or indeed than is generally supposed in GBM.

Does more sophisticated SNV filtering software increase or decrease the degree of similarity between replicates?

As an additional computational validation step for somatic mutations, it is common practice to employ computational algorithms that attempt to distinguish somatic mutations from germ line mutations and sequencing errors (Larson et al., 2012; Alioto et al., 2014). Software platforms to perform these tasks are plentiful, and each one typically employs multiple methods to identify true positive mutations. The two platforms used in this research, SomaticSniper (Larson et al., 2012) and GATK (McKenna et al., 2010), calculate one or more quality scores based on features of the dataset and the individual reads, and putative mutations with higher quality scores are considered to be more likely true somatic mutations than those with low quality scores. In addition to considering these quality scores, there are additional filtering steps that one can use to distinguish true somatic mutations from errors of all sorts. Table 2 lists eight distinct SNV filters we evaluated. The first three are based on the quality scores generated by GATK and SomaticSniper. We simply remove from the dataset anything that fails to meet these conventional quality-score thresholds. The other five are additional filters that we developed for this project. Each of these five filters represents an aspect of the data or the putative mutation that is generally thought to indicate that a given SNV call is a false positive.

We first asked whether filtering the data increases or decreases the percentage of the sample that is overlapping between the two technical replicates. We expect that this analysis informs us about whether filtering out putative somatic mutations with these features affects the proportion of true somatic mutations in the remaining dataset. We found that, after removing putative mutations tagged by any one of the eight filters, the number of putative mutations per replicate decreased from 23–10,966 to 0–14 (Table 1). The size of the overlap between technical replicates decreased to 0–2 per sample, with 0 as the mode, i.e., the overall overlap percentage also decreased. We concluded that running all the filters on the data, in the absence of any other verification method, was counterproductive, because it removed all of the signal (as well as all the noise).

We next looked at the individual effects of six of eight filters (We did not specifically study the GATK and SS filters, since both are directly linked to read quality.). We ran each filter independently on the whole dataset to see which putative SNVs it caught. We first considered the total number of putative SNVs removed by each filter (Fig. 4) on each sample. We found that different filters removed different numbers of mutations, and that the lion’s share of mutations were removed by the VAQ (Variant Allele Quality) and LOH (Loss of Heterozygosity) filters, which removed on average 309 and 539 putative SNVs, respectively (Table 1). Three other filters, those removing overlap with dbSNP and mutations within a 10 bp window of indels or other SNVs, removed on average 16, 28, and 46 putative mutations per sample, respectively (Fig. 4 and Table 1). The final filter, which removed putative SNVs with less than 10% coverage of the alternate allele, removed 1 putative SNV on average.

Figure 4 Effect of individual filters on the overlap between replicates.

For each filter (names given on the x-axis, with detailed description in Table 1), we looked at (A) the number of mutations removed by a given filter in a given sample on a log scale and (B) the percentage of the WGS–WGA overlap removed by the filter, per sample. The LOH and VAQ filters removed a large number of putative SNVs and portion of the overlap. The <10% filter removed very few putative SNVs and almost none of the overlap. The 10bp-SNV, 10bp-INDEL, and dbSNP filters removed an intermediate number of putative SNVs (100), but only a very small portion of the overlap, making them the best performers on the overlap data.

We next looked at the individual effects of the six of eight filters on the overlap between replicates. We asked how many of the SNVs removed by each filter were SNVs present in the overlap between technical replicates, and how many were in just one sample? Put differently, what percentage of putative SNVs removed by a given filter was in the category more likely to be true positives (overlap), versus the category more likely to be false positives (only present in one replicate)? To answer this question, we plotted the percent of the overlap (per sample) that was removed by each of the six filters (Fig. 4). We found that the three filters removing overlap with dbSNP and putative mutations within 10bp of an indel or another SNV removed, on average, only 3%–4% of the overlap (Table 1). By contrast, the VAQ filter (specific to SomaticSniper) and the LOH filter each removed 53% and 51% of the overlap, respectively (Fig. 4 and Table 1). Thus, our evidence suggests that the filters removing overlap with dbSNP and putative mutations near other putative mutations are preferentially removing false positives, while the filters removing low VAQ and LOH are less discriminatory and may be removing a large proportion of the true positives.

Figure 5 Effect of individual fiters on the difference between replicates.

For each sample, we compared the size of the difference between replicates (the number of SNVs recovered from only one of the WGA and WGS replicates). (A) A scatter plot of the ratio, per sample per filter, of putative SNVs removed from the difference versus the overlap of WGS and WGA (WGS Δ WGA/WGS ∩ WGA). (B) Boxplot of the same data, divided by filter on the x-axis. (C, D) Plot of the percentage difference in the Jaccard (WGS, WGA), with Jaccard = |WGS ∩ WGA|/|WGS ∪ WGA|, before and after the action of each filter (each filter was run on the whole data set independently).

Next, we looked at the individual effects of the six of eight filters on the difference between replicates of each sample: those mutations found only in WGS or WGA, but not both. We looked at the ratio, per sample per filter, of putative SNVs removed from the difference versus the overlap of WGS and WGA (WGS△WGA∕WGS∩WGA), where a higher ratio indicates that more mutations are removed from the difference than the overlap. We first plotted this ratio for each sample for each filter (Fig. 5A). To better compare the filters to each other, we looked at the distribution of ratios across samples for each filter (Fig. 5B). We found that the LOH and VAQ filters scored worst, followed by the 10bp-SNV, 10bp-INDEL, and dbSNP filters, in that order (We did not look at the <10% filter for this analysis, because it did not catch enough data to be meaningful.). Thus, those filters that remove the fewest putative SNVs in the overlap, also remove the highest number of SNVs in the difference relative to the overlap.

We next looked at the similarity of WGS to WGA as a whole as measured by the difference, normalized to 100, in the Jaccard similarity coefficient (WGS∩WGA∕WGS∪WGA) before and after filtering (Jaccard after-Jaccard before /Jaccard before). By this measure, and in contrast to the previous two measures, we found that LOH and VAQ performed the best by an order of magnitude, followed by 10 bp-SNV, 10bp-INDEL, and dbSP, in that order (Fig. 5, below). We see this effect because these two filters remove so much data: half or more of the overlap and as much or more of the difference. Removing so much of the data makes the intersection much, much smaller, and thus makes the Jaccard coefficient between samples much larger, mostly independently of the size of the overlap.

Figure 6 LOH and VAQ filters remove almost completely different sets of mutations and cover most of the sample.

Across samples, as the length of the overlap between WGS and WGA increases (x-axis), (A) the percentage of the overlap filtered out by LOH increases, and (B) the percentage of the overlap filtered out by VAQ decreases. The two are almost (but not quite) perfect inverses: putative SNVs filtered by LOH and VAQ cover almost the entire overlap, and together sum to nearly 100% of the overlap.

Finally, we asked whether either the VAQ or the LOH filter, responsible between them for removing most to all of the overlap, was more likely to remove overlap for samples that showed a lot of overlap. We plotted the percent of the overlap filtered out by the LOH filter (Fig. 6) and the percent of the overlap filtered out by the VAQ filter (Fig. 6) against the number of putative mutations in the overlap, and we found that for samples with overlap of ≲100 there was no strong trend. Either filter was removing between 0 and 100% of the overlap for some samples. For samples with more overlap, however, LOH was the primary filter removing overlap.

We found an additional trend that was initially unexpected: the LOH and VAQ graphs (Fig. 6) looked like exact inverses of each other. Further inspection showed that the sum of the percent overlap removed by LOH and by VAQ was nearly, but not exactly, 100% in all cases. In hindsight, this result was somewhat expected, since (i) each of the LOH and VAQ filters removes about half of the overlap, and (ii) all or almost all of the overlap is removed every time.

Discussion

GBM is an evolutionary disease that develops when mutations arise in glial cell lines and the mutated cells and their lineages subsequently co-opt the surrounding tissue and systems to the detriment of the organism as a whole. Treatment for GBM is difficult and has poor outcomes (Wilson et al., 2014), but may be improved by a more complete understanding of the somatic mutations present in GBM. Large-scale sequencing projects, like TCGA, make cancer sequencing data available to many researchers. These data are of enormous potential value to the research community, but their accuracy and reproducibility are unknown. Here, we have made a first step towards evaluating the reliability of the TCGA data, by comparing 55 technical replicates in the TCGA GBM dataset.

We found that, on average, about half of the putative mutations in the raw data for the WGS replicate (no amplification before library preparation) and about a third of those in the WGA replicate (with amplification before library preparation) were present in both replicates. The number of mutations present in both replicates was anywhere between 20 and 5,000 putative mutations. We found further that the high number of putative somatic mutations in some, but not all, of the patient samples was repeatable across technical replicates. Moreover, samples with a higher frequency of putative mutations had equally similar technical replicates to those samples with a lower frequency of putative mutations. These results suggest the possibility that a higher mutation frequency could be a feature of a subset of GBM tumors and not a data artifact.

Filtering the raw computational data using both quality scores from GATK and SomaticSniper, as well as five additional custom filters, eliminated more than half of the total number of putative mutations in all 110 samples, including most or all of those present in both replicates. To some extent, this result was unsurprising: these filters were designed to be used on wild type genomes, where it is generally assumed that any observed differences are more likely due to error than to the presence of true SNVs. For example, LOH in this case is more likely to be an error than an actual mutation. When we do cancer genomics, however, our goal is the opposite, to highlight changes. Therefore, it is possible that we need entirely different filtering protocols. There are also more theoretical reasons to consider altering the filtering protocols for cancer genomes. For example, multiple sources suggest that LOH mutations may be essential to cancer (Fujimoto et al., 1989). This hypothesis, along with the data presented here, makes a strong argument for retaining these mutations in functional analyses rather than excluding them.

Of the six filters whose individual effects we examined, only two, those that removed Loss of Heterozygosity (LOH) mutations and putative mutations with low VAQ (calculated by SomaticSniper), removed primarily mutations that we found in both technical replicates. In combination, they removed most or all of those mutations present in both replicates, since the two filters consistently removed nearly completely disjoint sets of putative mutations. Of the remaining four filters, only three removed any appreciable number of putative mutations from the sample, and each of these preferentially removed mutations present in only one technical replicate. Two of these three filters, those removing putative SNVs within a 10 bp window of putative indels or other putative SNVs, recognize a feature (clustered mutations) that suggests a local problem with the reads or alignment. The third removes overlap with dbGaP. Our analysis suggests that these three filters do clean up the data in a meaningful way. By contrast, it may be more useful not to apply the two filters that removed principally data from the overlap of the two replicates.

Several factors limit the conclusions we may draw from this analysis. First, in this analysis we used repeatability between technical replicates (being in the WGS and WGA samples) as a measure of confidence in a putative SNV. This metric is potentially problematic for many reasons, including but not limited to: (i) cancer is highly heterogeneous, and so a legitimate somatic SNV might show up in one replicate but not another; (ii) if the DNA sample is degraded to some extent, due to surgery conditions or some other factor out of the hands of the sequencing center, the same errors may appear in both replicates; (iii) the SNV calling process may enrich for artifacts such as germ line variants, which are nearly indistinguishable from somatic SNVs in computational analyses; cross-referencing to gold standards such as dbSNP is the only way to identify germline SNPs; (iv) a putative mutation that is present in both samples may be a somatic mutation that arose before the tumor (Tomasettia et al., 2013). Although repeatability across technical replicates cannot guarantee that a putative mutation is a true somatic mutation, it does increase the likelihood. Having an independent measure of confidence in SNV calls, even an imperfect one, can help us gauge the accuracy of other measures, specifically of different filtering approaches.

Second, we studied only one particular SNV caller, SomaticSniper (Larson et al., 2012). There are other, equally widely used SNV callers available (Cibulskis et al., 2013; Koboldt et al., 2009; Saunders et al., 2012), which might do better or worse than the one we have chosen here. Also, we did not even consider every possible quality metric available in the SNV-caller we did choose. For example, we did not look at the effects of the SomaticScore or the GATK quality score individually. In the future, it would be worthwhile to evaluate other SNV-callers and other quality metrics.

We have shown that there is significant overlap between technical replicates of whole exome sequencing in the TCGA GBM dataset, comprising about 50% of putative SNVs in WGS samples and about 30% in WGA samples. The overlap exists even for samples with a high number of putative SNVs, suggesting that some GBMs may have significantly more somatic mutation than others. We acknowledge that the high rate of non-concordance between replicates in our analyses indicates that even the best computational analyses are insufficient to validate any single SNV; when the identity of a single SNV in a single sample is important, validation by an orthogonal method (e.g., PCR, Sanger sequencing) remains necessary. Nevertheless, when less fine-tuned methods are acceptable, or are the only option available, one may wish to employ other methods of validation rather than nothing, such as the six data filters that are commonly applied to validate SNVs that we examined. We found that some of these filters remove principally those mutations found in one sample or the other, while other filers remove primarily those in the overlap. We suggest that when orthogonal validation is not an option, only the filters that removed little overlap between samples should be used for computational SNV validation.

Supplemental Information

Figure S1 One third (WGA) to one half (WGS) of putative SNVs were recovered in technical replicates

For each pair of replicates (WGS and WGA), we looked at the percentage of WGS SNVs that were recovered in the WGA sample (about one half, in green), and the percentage of the WGA SNVs that were recovered in the WGS sample (about one third, in orange). The WGS distribution is higher and narrower, showing that the WGS samples overall have a higher percentage overlap than the WGA samples, and less range in this parameter.

Click here for additional data file.

Figure S2 Number of putative SNVs in a sample does not correlate with coverage

The number of SNVs called in a sample does not correlate with the coverage of that sample (Spearman ρ = − 0.13, S = 671817, P = 0.12). This is shown by the consistent variability along the x-axis at each level of overage (shown on the y-axis). The separation of the two experimental condition on the y-axis is not relevant to this measure.

Click here for additional data file.

Figure S3 Number of putative SNVs in a sample does not correlate with percentage of mapped reads

The number of SNVs called in a sample does not correlate with the percentage of mapped reads in the alignment of that sample (Spearman ρ = − 0.068, S = 637326, P = 0.41).

Click here for additional data file.

Table S1 Back-end Processing (PDF)

This table shows the software packages we used in data processing, what we used each piece of software for, and the command associated with the use of it. The rows are in order of use.

Click here for additional data file.

Supplemental Information 1 Back-end Processing (TEX)

This table shows the software packages we used in data processing, what we used each piece of software for, and the command associated with it. The rows are in order of use.

Click here for additional data file.

Additional Information and Declarations

Competing Interests

Author Contributions

Data Availability

Claus Wilke is an Academic Editor for PeerJ.

Dakota Z. Derryberry conceived and designed the experiments, performed the experiments, analyzed the data, contributed reagents/materials/analysis tools, wrote the paper, prepared figures and/or tables, reviewed drafts of the paper.

Matthew C. Cowperthwaite conceived and designed the experiments, contributed reagents/materials/analysis tools, wrote the paper, reviewed drafts of the paper.

Claus O. Wilke conceived and designed the experiments, wrote the paper, reviewed drafts of the paper.

The following information was supplied regarding data availability:

https://github.com/clauswilke/GBM˙genomics.

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
