# Peer review of "Reproducibility of SNV-calling in multiple sequencing runs from single tumors"

_PeerJ, doi:10.7717/peerj.1508_

## Round 0.1 · original submission · Minor Revisions

The two reviewers raise some common questions such as the redundant information in the figures. It would be best to follow the reviewer's suggestion to revise and re-organize the figures.

Reviewer 1 ·

Basic reporting

The article is written in English using clear and unambiguous text.

The article include sufficient introduction and background.

The structure of the article contains Title, Abstract, Introduction, Results, Discussion, and Methods, which almost conform to PeerJ standard sections (https://peerj.com/about/author-instructions/). The affiliations of authors need to be rearranged, as department, university, location, country, and contact information of corresponding author.

Figures are relevant to the content of the article. The font size in Fig. 1 is small. And some axis labels are confusing. 'No.' in Fig. 2 and Fig. 4 is better to be writtern as 'Number' or '#'. The x-axis label '% of sample obtained in overlap' of Fig. 3 is confusing. The x-axis label 'Length of overlap' in Fig. 7 and Fig. 8 is better to be 'Number of overlap'. And accordance between figures should be double-checked, including whether the first latter should be capitalized, usage of 'number' or '#', and 'percent' or '%'.

The work is 'self-contained', although some terminology looks not well-defined, and confusing, such as 'polymorphic mutation' and 'mutation frequency'.

Their raw data is downloaded from TCGA, and the authors claimed that the analyzing python and R scripts are available in a public git repository (github.com/clauswilke/GBM_genomics), which I attempted to open, but failed with '404 Page not found'.

Experimental design

The submission is within the scope of PeerJ.

The authors tried to assess the quality of TCGA sequencing data and somatic SNV-calling, by applying a general somatic SNV calling pipeline and several filters to technical replicates of 55 GBM samples, and analyzing the size of overlap.

The analysis is a good attempt, which is, however, quite rough. The details are put in the 'General Comments for the Author' part.

The methods look probably reproducible.

Validity of the findings

The authors used public data of technical replicates from TCGA.

Their analysis may need to be improved. And the conclusions and discussions may need revision.

Comments for the author

Page1-2, Abstract and Introduction. When I read the Results, I suddenly recognized that the authors seemed to talk about somatic mutations by comparing tumor and blood, and figure out the overlap of called somatic mutations between two technical replicates. If this is what they mean, I think it is better to emphasize somatic mutation or somatic SNV in Abstract and Introduction.

Also, in Fig. 1, it is unclear what is the use of the blood sample. So the diagram is better to be redrawn to show that.

Page 3, Results. The authors seem to use the phrase 'polymorphic mutation' to mean a somatic mutation site. I am not sure whether 'polymorphic mutation' is a correct, commonly-used term.

Page 3, Results. The authors said 'Theoretically, any fixed mutation will appear in all replicates, while errors due to (i) sequencing errors, (ii) amplification errors, or (iii) alignment errors will not.' However, in my opinion, system errors in alignment and variant-calling can actually appear in replicates. Although the authors discussed the errors in both replicates in Discussion, such as DNA degradation and sequencing center factor, they did not mention any system errors due to alignment and variant-calling, which I think might dampen their analysis, and must be analyzed or discussed. In my experience, searching for somatic mutations will enrich more such artifacts than for germline variatns, and thus, PCR is required to validate the putative mutations. Therefore, I will not regard the overlap as true positives, although I believe true positives will be in the overlap, if they exist.

For Fig. 2 and Fig. 4 and related analysis, I am not sure how the authors computed the Pearson correlation coefficient, whether it is computed on the original count number scale or on the log scale. (According to the distribution, I think the latter is more reasonable). Computing on different scale may give different Pearson correlation coefficient. Therefore, I suggest to use Spearman correlation coefficient and corresponding test, which will not change on different scale.

Page 4. The authors said 'We further found that the percent overlap between the two samples, calculated as ..., was fairly consistent, on average 31% in WGA replicates and 44% in WGS replicates (Table 1).' However, in Table 1, it is clear to see that the percent overlap ranged from 1-74% for WGS (avg. 31%, sd 20%) and 3-71% for WGA (avg. 44%, sd 13%). Can such huge variation range be said 'fairly consistent'? And the average percent is not consistent between the main text in Page 4 and the number in Table 1.

For Fig. 3, I am not sure whether Fig. 3 is meaningful after given Fig. 2. Because the number of putative SNVs is larger in WGA than in WGS, the overlap proportion is certainly higher in WGS than WGA. And as shown in Fig. 2 and Table 1, the variance of the number of putative SNVs is also larger in WGA than in WGS, so it might be reasonable to see the variance of proportion of overlap is larger in WGA than in WGS. Thus, the meaning of Fig. 3 seems to be vague, and the sentence describing Fig. 3 in the text seems not to provide any more information. It might be better to use a scatter plot, showing the proportion of overlap against number of putative SNVs in WGA (just like Fig. 4, proportion of overlap against number of putative SNVs in WGS), which may merged with Fig. 4 with different dot color or shape, and segments or arrows connecting corresponding dots on the merged figure may be added.

Page 5-6. 'Does more sophisticated SNV filtering software increase or decrease the degree of similarity between replicates?' In this part, the authors analyzed the influence of several filters on the number of putative mutations and the overlap between techinical replicates. They focused on the overall number of filtered putative mutations and the percent of filtered among overlap, and correspondingly plotted Fig. 5-8. In Page 6, they mentioned to answer 'what percentage of putative SNVs removed by a given filter was in the category more likely to be true positives (overlap), versus the category more likely to be false positives (only present in one replicate)?'. However, they only demenstrated some filters removed many putative sites and corresponding much percentage among overlap. They did not directly compare the percentage between the overlap and the list only presenting in one replicate (I calls it 'diffset' in the following), or whether each filters removed more proportion of false positives than true positives or vice versa, which I think should be done. This might be done by comparing the Jaccard similarity coefficient (= overlap / union) before and after each filter, or doing Fisher's exact test on two-by-two contingency table of how many sites were filtered/not filtered for overlap/diffset.
And thus, Table 1 might be too summary for futher examining and explaining their results. I suggest to make a supplementary table containing the sample name and the statistics in Table 1 of amount (proportion is not necessary) for each of 55*2=110 tumor samples.

Besides, as shown in Fig. 7 and Fig. 8, the two figures looked like mirror, which is quite interesing. And as mentioned in the last paragraph of Results, the authors' explanation implied that the percentage was calculated when filtering putative sites only once using all six filters sequentially. If this is true, the order of the six filters may affect the results, and Fig. 7 and Fig. 8 might not be so meaningful. In fact, I think it is better to evaluate each filter on the same original putative sites, and report their effect on overlap and also on diffset. And I am not sure why the authors said '... we found that for samples with overlap of < 100 there was no strong trend.' (why the authors focused on overlap number < 100), for Fig. 7 seems to have a clear positive correlation trend and Fig. 8 with a clear negative correlation trend.

By the way, as shown in Table 1, each statistic seemed to vary much between samples. And the authors reported that the VAQ filter removed a large proportion of putative mutations, as shown in Fig. 8. So I wonder if these 110 samples of GBM in TCGA have much variation or 'heterogeneity' on their sequencing quality, so that the putative mutations from the samples with very low quality will probably be filtered. Therefore, I think the authors may need to describe more about the data and their quality (homogeneity).

In summary, I list the two most important problems: (1) overlap does not mean all true positives, thus filtering in overlap might not always be wrong, so it might be more clear to compare the filtering effect in overlap and diffset; (2) filters may need to be evaluated separately, and if Fig. 7 and Fig. 8 are still in that mirror shape, it will be interesting.

Reviewer 2 ·

Basic reporting

No Comments

Experimental design

No Comments

Validity of the findings

No Comments

Comments for the author

It's a very comprehensive study on QC in WGS analysis. I concern on two points:

1. There are too many figures and tables in the maintext. It seems a little bit duplication among some of them. For example, both table3 and figure 1 are focused on data processing process. The workflow in figure is clear enough for reader to get the whole experimental design. The Table3 which contains the technical details of the workflow could be move to supplementary.

2. Generally, the number of SNVs in WGA are larger than WGS. The author mentioned that it may caused by the greater number of amplification errors. Herein, I'm interested on the percentage of mapped reads among the bam files of WGS and WGA groups after local alignment process in GATK. The higher percentage of mapped reads may also indicate more SNVs. If the mapped reads are in the similar level of WGA and WGS, it would be more reliable to conclude that the larger number of SNVs was caused by amplification errors.

---

## Round 0.2 · Minor Revisions

Please improve the figure and figure legend as suggested by reviewer 1.

Reviewer 1 ·

Basic reporting

The article is written in English using clear and unambiguous text.

Experimental design

The submission is within the scope of PeerJ.

Validity of the findings

The authors used public Glioblastoma multiformed (GBM) data set with technical replicates from TCGA.

Comments for the author

The manuscript looks much better now, and I think it is almost acceptable.

Here are some little suggestions:

1) Double-check the index of figures and tables. For example, on page 33 and page 35 of the 'peerj_reveiwing_*.pdf' I believe the title should be Figure 5 and Figure 6 instead of Table 3 and Figure 5, which I am not sure if is due to PeerJ review system.

2) Table S1 could not be downloaded from PeerJ, which seeemed to be in .tex format. It might be better provided in .xls or .pdf format.

3) The color (hue) for WGA and WGS is better to be the same or similar (accordant) in relavent figures, such as Figure 1, Figure 3 and Figure S1.

4) To better answer previous Reviewer 2's comment (2), it might be better to distinguish WGA and WGS in Figures S2 and S3, such as in different color.

5) In Figure 5, as the top two subplot represent the same data, I think their y-axis should be in the same scale (0-80) and aligned. For better clarity, the y-axis labels might be needed for all four subplot, and it might be better added the formula of delta_Jaccard to the figure legend.

6) In Methods and legend of Figures S2 and S3, the mean, stdev, spearman rho and p-value seem to have too many significant digits.

7) The legend in Figure 3 is better moved to top left, so that it will not overlap a data point.

8) Adjust the x-axis label of Figure S1, so that it could be totally seen.

9) For Figure 3, the scatter points from WGA and WGS seemed to be closely distributed, so it might be better to use different symbols for them, such as circle and cross.

10) Double-check the figures and tables, including their legend, especially for supplementary materials. They are better to be self-contained.

Reviewer 3 ·

Basic reporting

No Comments

Experimental design

No Comments

Validity of the findings

No Comments

---

## Round 0.3 · accepted · Accept

Please double check the format and figure quality for PeerJ publication.